# GLP1 Receptor Agonists—Effects beyond Obesity and Diabetes

**DOI:** 10.3390/cells13010065

**Published:** 2023-12-28

**Authors:** Sydney S. Wilbon, Mikhail G. Kolonin

**Affiliations:** The Brown Foundation Institute of Molecular Medicine, University of Texas Health Science Center, Houston, TX 77030, USA; sydney.s.wilbon@uth.tmc.edu

**Keywords:** GLP-1 receptor agonist, aging, cancer, diabetes, incretin, obesity, neurodegenerative disease, cardiovascular disease

## Abstract

Glucagon-like peptide-1 receptor agonists (GLP1RA) have been transformative for patients and clinicians in treating type-2 diabetes and obesity. Drugs of this class, the bioavailability of which is continuously improving, enable weight loss and control blood glucose with minimal unwanted side effects. Since adopting GLP1RA for treating metabolic diseases, animal and clinical studies have revealed their beneficial effects on several other pathologies, including cardiovascular diseases, neurodegeneration, kidney disease, and cancer. A notable commonality between these diseases is their association with older age. Clinical trials and preclinical data suggest that GLP1RA may improve outcomes in these aging-related diseases. Some of the benefits of GLP1RA may be indirect due to their effects on obesity and glucose metabolism. However, there is building evidence that GLP1RA may also act directly on multiple organs implicated in aging-related pathology. This review aims to compile the studies reporting the effects of GLP1RA on aging-related diseases and discuss potential underlying mechanisms.

## 1. Introduction

Diabetes affects over 11% of the U.S. population and is currently the seventh leading cause of death. Aging and obesity are the main risk factors for type 2 diabetes (T2D) development, with about 40% of the diabetic population in the U.S. being over the age of 65 [1]. The leading causes of death in these patients with T2D are cardiovascular diseases and cancer due to their pathogenesis being aggravated by metabolic syndrome [2]. The recently approved glucagon-like peptide-1 receptor agonists (GLP1RA) have been groundbreaking in their ability to treat diabetes and obesity with a comparatively benign side effect profile. Despite the widespread adoption of these drugs, the extent of their therapeutic value has not fully been elucidated. These drugs were formulated to function like the endogenous glucagon-like peptide 1 (GLP1), an incretin hormone produced by L cells in the intestine. Postprandially, the levels of GLP1 increase dramatically to stimulate insulin biosynthesis and release, suppress glucagon release, delay gastric emptying, improve glucose metabolism, and increase satiety [3]. Driven by the discovery of the metabolic effects of GLP1 and its fragment exendin-4 (exenatide), pharmaceutical companies have developed improved GLP1 analogs for metronomic administration and sustained release. Several of these including liraglutide (Victoza, Saxenda), dulaglutide (Trulicity), and semaglutide (Ozempic, Rybelsus, and Wegovy) have been FDA-approved [4,5]. While GLP1RA had been initially indicated for T2D, liraglutide, and semaglutide have been subsequently approved for the management of obesity. Recently, semaglutide has also been approved to reduce the risk of cardiovascular events in patients with T2D and confirmed cardiovascular disease [6]. In addition, these agents have been noticed to reduce renal dysfunction [7] and liver steatosis [8]. 

The GLP1R is expressed and can be activated in various cells of the pancreas, digestive tract, heart muscle, skeletal muscle, liver, central nervous system, and immune system [9,10,11]. Given the multiple sites of GLP1R expression, many of its effects across the many tissues expressing it are yet to be fully understood. For example, clinical trials have shown that GLP1RA treatment to be protective against skeletal muscle wasting (Table 1). The risk of sarcopenia, T2D, cardiovascular disease, neurodegenerative disease, cancer, and renal disease is positively correlated with biological aging. The purpose of this review is to compile information on the systemic effects of GLP1RAs that could mediate their anti-aging effects via their activities in multiple organs (Figure 1). The synthesis of data on GLP1RA’s effects on pathways underlying the aging of vital organs provides new insights into how these therapeutics mitigate aging-associated diseases.

## 2. GLP1RA Effects on Diabetes

The GLP1RA have been proven across multiple clinical trials to improve glucose control in patients with diabetes. In clinical studies, liraglutide has been shown to increase pancreatic β cell function, as shown by improved glucose tolerance and a corresponding decrease in HbA1C levels. GLP1RA have been found to increase glucose-stimulated C-peptide secretion, a surrogate for insulin secretion. Notably, GLP1RA-treated patients consistently have a significant decrease in their BMI [18]. Cessation of the drug most often causes a reversal of the beneficial effects [18]. However, there is evidence that these drugs may have long-term effects. In a study of patients taking exenatide for three years, patients had improved glucose-stimulated C-peptide secretion four weeks after drug cessation. Although this post-withdrawal effect did not translate to improved glycemic control, it does show that some beneficial alterations in β cell function are preserved with more long-term treatment [19].

There are several possible reasons for the improvement in glucose metabolism with the use of GLP1RA, but their effects on pancreatic β cells are of great significance. When tested in vivo in the genetic mouse model of obesity, exenatide decreased β cell endoplasmic reticulum (ER) stress signaling in pancreatic islets. Ex vivo, exenatide increased insulin production and protected isolated β cells from chemically induced apoptosis in a PKA-dependent manner [20]. In another study, infusion of GLP1 rescued rats from aging-related declines in insulin secretion and glucose tolerance. GLP1 also caused increased pancreatic mRNA expression of *glucokinase* and *glut2* [21]. GLP1RA have also been shown to improve hyperglycemia by increasing β cell mass. Exenatide and GLP1 both were able to increase pancreatic expression of IDX-1, a transcription factor mediating β cell expansion [22]. These studies highlight the ability of GLP1RA to act as disease-modifying drugs in T2D. Long-term effects of incretin mimetics in other organs, including the immune system, adipose tissue (AT), liver, and skeletal muscle, have been proposed to contribute to their sustained anti-diabetic effects [23] (Figure 1). 

## 3. GLP1RA Effects on Obesity

The prevalence of obesity in the U.S. has increased from 30.5% in 2000 to 41.9% in 2020 [24]. Although obesity rates are relatively stable across age groups, the metabolic derangements caused by obesity are cumulative over the lifespan. These abnormalities can progress to metabolic syndrome, characterized by hyperglycemia, hypertension, and dyslipidemia. There is increasing evidence that obesity and aging dysregulate many overlapping signaling pathways, including inflammation and oxidative stress, and that obesity accelerates the onset of aging-related diseases. In support of this, obese patients have decreased lifespans and poorer clinical outcomes compared to their lean counterparts. Obesity is a condition of AT overgrowth most often caused by an imbalance between caloric intake and expenditure [25]. AT is a heterogeneous metabolic and endocrine organ that is key in regulating energy homeostasis. It is prominent subcutaneously and around visceral organs but also exists in other metabolically distinct depots in the body. White adipose tissue (WAT) predominantly stores lipids for rapid mobilization, and brown adipose tissue (BAT) is responsible for non-shivering heat production [25]. Given the connection between obesity and aging, mechanisms by which GLP1RA affect AT signaling may provide new clues on their systemic physiological effects.

Semaglutide and liraglutide are the two GLP1RAs that have been FDA-approved to treat obesity. In a double-blinded, randomized control trial of type 2 diabetic patients, semaglutide treatment caused, on average, an extra 2.7 kg of body weight loss compared to patients on standard-of-care treatment [26]. Several other clinical trials have shown the ability of semaglutide to cause significant bodyweight loss [27,28]. Although not as effective as semaglutide, liraglutide also causes significant weight loss compared to placebo-treated patients [28,29]. There is also a novel non-peptide GLP1RA, orforglipron, that causes significant weight loss in overweight and obese patients, but it is not yet FDA-approved [30]. GLP1R agonism has confirmed benefits in obesity, but the mechanisms behind these changes are still being studied.

AT mass control has been attributed, in part, to the effects of GLP1RA on the sympathetic nervous system (SNS) and appetite suppression [31]. The “inducible/recruitable” brown-like (beige) adipocytes arise in subcutaneous AT (SAT) in response to catecholamine activation of adrenergic receptors. Receptor activation dramatically increases the metabolism of both glucose and fatty acids, thereby partially counteracting the metabolic consequences of obesity [32,33]. Thermogenic AT activation by prolonged GLP1 analog treatment has been reported for both mice and humans [34,35]. We recently reported that liraglutide treatment transiently induced interleukin-6 (IL6) in circulation and AT beiging in prediabetic patients and that this effect was recapitulated in the mouse model of obesity. The GLP1R-IL6 receptor signaling axis was shown to activate adipocyte lipolysis and thermogenesis [36]. Recombinant GLP1, beinaglutide, was also found to induce WAT beiging linked with its anti-obesity effects [37]. IL6 receptor knockout mice lack the increase in adipocyte beiging, energy expenditure, and glucose clearance in response to liraglutide treatment, demonstrating the GLP1RA signal through IL6 to increase adipocyte metabolism [36]. In addition, GLP1 and liraglutide have both been shown to increase the number of proliferating pre-adipocytes and induce differentiation of adipocytes [38]. Adipocyte hyperplasia, instead of hypertrophy, in response to lipid excess, is protective against insulin resistance and ectopic fat deposition in other organs [38]. Therefore, GLP1RA may protect from obesity and its consequences at least in part by increasing adipocyte hyperplasia and inducing AT beiging to increase energy expenditure. Notably, the knockout of the IL6 receptor in adipose cells did not negate the anti-obesity effect of liraglutide, indicating that GLP1RA signaling in the CNS may be more important for weight loss [36]. Future studies will be needed to establish if weight loss caused by GLP1RA is solely due to reduced food intake or whether GLP1RA signaling in AT contributes to their anti-obesity effect (Figure 1).

## 4. GLP1RA Effects on Cardiovascular Disease

Cardiovascular disease (CVD) is a significant cause of morbidity and mortality worldwide and is the number one cause of death in the U.S. [39]. CVD has the highest prevalence in the aged population, with incidences increasing from ~5% in those 18–44 to almost 35% in the 65+ population. Despite there being effective interventions to prevent and treat CVD, the rate in the 65+ population only saw a 4% decrease in incidence between 2008 and 2018 [40]. Molecular mechanisms of aging underlying the incidence and progression of CVD have not been sufficiently explored as a potential treatment target.

GLP1RA have shown efficacy and promise in the context of CVD. The results from various clinical studies show the multifactorial effects GLP1RA have on the cardiovascular system (Table 2). A meta-analysis of trials testing multiple GLP1RA in diabetic patients revealed that major adverse cardiovascular events, including stroke, myocardial infarction, and cardiovascular death, decreased by 12% in treated patients irrespective of the drug used or the population studied. Patients taking GLP1RA also experienced a 9% reduction in hospitalizations due to heart failure [41]. Interestingly, a rare missense mutation in the GLP1R appears to confer a reduced risk for coronary heart disease development [42]. In a study of diabetic patients, exenatide was found to transiently suppress the expression of inflammatory markers TNFα and IL-1β. In this study, treatment for 12 weeks reduced the circulating levels of proinflammatory mediators like MCP-1, serum amyloid A, and IL6. Notably, these long-term effects of chronically high and pathogenic IL6 are distinct from the short-term beneficial effects of GLP1RA on IL6 secretion [36]. The long-term effects of GLP1RA were also independent of weight loss [43]. Because much of the clinical trial data is limited to patients with diabetes, obesity, or both, it is possible that the disease-modifying effects of GLP1RA are due to the mitigation of metabolic abnormalities associated with these conditions. However, results from preclinical studies [44,45,46,47] suggest that at least some of the benefits are due to a direct effect of GLP1RA in the cardiovascular system (Figure 1). Future studies in aging lean and non-diabetic patients may help to determine how much GLP1RA directly benefits cardiovascular diseases and contribute to clinical outcomes.

In support of the clinical data, the mechanisms behind the beneficial effects of GLP1RA on CVD have begun to surface. Liraglutide has been shown to ameliorate inflammatory signaling in the atria of infarcted mice, induce transcription of structural proteins, and protect cells from ER stress by upregulating heat shock proteins. In this model, knockout of the GLP1R in endothelial cells extinguished the beneficial effects of liraglutide, as shown by diminished ejection fraction and larger infarct size compared to mice with intact GLP1R expression [44]. Consistent with these results, exenatide increased contractility of atria isolated from humans. Inhibitors of calcium release and the cAMP/PKA pathway blunted the positive ionotropic effect of exenatide, implying that the GLP1RA signal through this pathway improves atrial function [45]. In addition to endothelial cells, vascular smooth muscle cells (VSMC) play a pivotal role in the health of the cardiovascular system. In accordance with what has been shown in endothelial cells, exenatide was found to reduce angiotensin II-mediated VSMC senescence by reducing oxidative stress [47]. Exenatide has also been shown to decrease inflammatory signaling and the conversion of macrophages challenged with ox-LDL into foam cells, a culprit in atherosclerotic disease [46]. These preclinical studies highlight some of the direct effects of GLP1RA on CVD pathogenesis and progression.

## 5. GLP1RA Effects on Neurodegenerative Diseases

Neurodegenerative diseases are a group of disorders of the nervous system. Alzheimer’s disease (AD), the most common of them, is a progressive dementia characterized by the accumulation of misfolded beta-amyloid plaques and phosphorylated tau tangles in the brain. Age is the most significant risk factor for AD, with rates as high as 33% in the population older than 85; AD is the fifth leading cause of death in those 65 or older [56]. Large cohort autopsy studies have demonstrated that the presence of protein aggregates in the aging brain is independent of clinical features of neurodegenerative disease, indicating that aberrant protein metabolism is a hallmark of aging in general [57]. While therapies aimed at misfolded proteins may have some benefit, new safe and effective approaches to suppress neurodegeneration are awaited.

In a recent analysis of three double-blinded, placebo-controlled clinical trials of GLP1RA in diabetic patients, investigators discovered that patients in the treatment arm had a decrease in dementia incidence compared to placebo [58]. Moreover, GLP1RA-treated patients also had a reduced rate of dementia when compared to patients taking other anti-diabetic medications in a nationwide cohort of diabetic patients. Interestingly, these patients also had decreased mortality compared to patients who died without dementia, demonstrating a possible lifespan benefit with GLP1RA treatment compared to other anti-diabetic drugs [58]. In another randomized, placebo-controlled clinical trial in obese, prediabetic, and early diabetic patients testing the effect of liraglutide against a dietary and exercise intervention, patients who received liraglutide showed improvement in composite memory scores and exposure to GLP1RA was positively correlated with changes in memory scores. These benefits were seen despite comparable weight loss and glycemic control, indicating that the effect on cognitive function is independent of the effect on weight and hyperglycemia [59]. In the REWIND clinical trial, a double-blinded randomized control trial of diabetic patients with cardiovascular risks, dulaglutide reduced the probability of cognitive decline [60]. These trials highlight the ability of GLP1RA to suppress neurodegeneration associated with aging (Table 3).

Consistent with these encouraging clinical trial results, the possible mechanisms of GLP1RA action in the brain have been elucidated in preclinical studies. It has been shown that liraglutide improves performance in the novel object recognition test and water maze task in APP/PS1 mice, a mouse model of AD. Importantly, these mice also showed decreased plaque burden in the cortex and reduced microglial activation, indicative of decreased inflammation and improved proteostasis. Liraglutide treatment in this model also restored the proliferation of neurons in the dentate gyrus and increased the number of synapses compared to control mice [64]. Liraglutide also improved memory retention in the SAMP8 mouse, a mouse model of accelerated aging and sporadic AD. This model demonstrated a liraglutide-dependent increase in the number of CA1 pyramidal neurons in the hippocampus region implicated in human memory loss [65]. In an in vitro model of beta-amyloid toxicity in neurons, treatment with semaglutide reversed the negative effects of beta-amyloid on cell viability by increasing the expression of an anti-apoptotic effector, Bcl-2, and decreasing the expression of pro-apoptotic effector, Bax. Semaglutide-treated neurons also displayed autophagy restoration that was compromised by beta-amyloid treatment [66]. Likely, the ability of GLP1RA to mitigate obesity and hyperglycemia partly accounts for their effects on the CNS. However, the potential importance of direct effects of GLP1RA on the cells of the brain, possibly mediated by their anti-inflammatory effects, will need to be further investigated.

## 6. GLP1RA Effects on Kidney Disease

About 14% of the adult U.S. population has chronic kidney disease (CKD), the progressive decline in kidney function that can be due to several etiologies [67]. The health of the kidneys can be determined using the estimated glomerular filtration rate (eGFR). eGFR in healthy individuals aged 20–40 is about 107 mL/min/1.73 m^2^ and undergoes a natural decline of 0.7 mL/min/1.73 m^2^ per year. Because of this natural aging-related decline, most healthy adults will have a low eGFR of <60 mL/min/1.73 m^2^ by 75 years old, which merits the diagnosis of CKD. Although these otherwise healthy patients do not experience proteinuria or any major increase in their relative risk of all-cause mortality, the high prevalence of chronic diseases like diabetes, obesity, and hypertension in the aged population exacerbates this natural aging-related decline in eGFR [68]. With the overwhelming evidence of the beneficial effects of GLP1RA on diabetes and obesity, it is important to better understand their ability to improve renal outcomes in the aged population.

In a nationwide cohort study comparing diabetic patients taking long-acting insulin vs. GLP1RA, the use of GLP1RA was associated with a lower risk of CKD progression, dialysis initiation, and renal death [69]. This finding is in accordance with an earlier trial comparing insulin glargine and dulaglutide in diabetic patients with moderate to severe CKD. These authors found that dulaglutide-treated patients had a higher eGFR than those on insulin [70]. Dulaglutide has also been previously shown to decrease the occurrence of new-onset macroalbuminuria, and treated patients had lower urinary albumin to creatinine ratio on follow-up [71]. Interestingly, adjusting for the decrease in HbA1C in the treatment group attenuated the renal effects of dulaglutide by about 26%, suggesting that the efficacy of GLP1RA is independent of their ability to alter this major CKD risk factor [71]. A separate randomized, placebo-controlled clinical trial testing the not-yet-approved GLP1RA efpeglenatide demonstrated a significant reduction in urinary albumin/creatinine ratio and increased eGFR. These beneficial effects were significantly independent of SGLT2 inhibitors and metformin use, again consistent with the possibility that the GLP1RA mechanism of action is independent of the effects of the standard-of-care drugs [55]. In addition to previously reported clinical data, the recently undertaken FLOW clinical study, a double-blinded, randomized control trial evaluating the ability of semaglutide to reduce kidney outcomes in patients with T2D and eGFR between 50 and 75 mL/min/1.73 m^2^ was stopped early by an independent data monitoring committee due to meeting prespecified criteria for efficacy. The trial’s primary endpoints included time to first kidney failure (persistent eGFR < 15 mL/min/1.73 m^2^ or initiation of chronic kidney replacement therapy), persistent ≥ 50% reduction in eGFR, or death from kidney or CV causes [72,73]. This and previous clinical trials give evidence supporting the role of GLP1RA in treating CKD (Table 4). However, more research is needed to understand how these drugs function in the kidney.

The mechanism of action of GLP1RA on the renal system and, subsequently, the occurrence and progression of CKD appears to be multifactorial. One mechanism involves the ability of GLP1RA to reduce tubular damage and inflammation in the kidney. It has been reported that overexpression of GLP1 in the db/db mouse model of obesity reduces histological markers of tubulointerstitial damage and reduces renal mRNA expression of pro-inflammatory markers *Tnfa* and *Ccl5*. These expression changes were accompanied by decreased renal infiltration of CD3+ T cells and F4/80+ macrophages. GLP1-overexpressing mice also had a significant survival advantage [76]. Consistent with that, it has been reported that global GLP1R knockout causes spontaneous kidney injury characterized by albuminuria and glomerulosclerosis that is exacerbated in streptozotocin-treated mice, a model of type-1 diabetes. This study also demonstrated the ability of liraglutide to improve podocyte architecture, albuminuria, and glomerulosclerosis. These histological markers corresponded with decreased expression of the inflammatory mediator NF-κB, and monocyte chemoattractant MCP-1. Despite the lack of detectable change in glycemia, similar results were seen in the liraglutide-treated nephrectomized rats, a non-diabetic kidney disease model. This study highlights the importance of GLP1 activity in baseline kidney function and establishes the utility of GLP1RA even outside of diabetic kidney disease [77]. In addition to the beneficial roles of GLP1R activation on fibrosis and proteinuria, this pathway has also been shown to alter renal hemodynamics. GLP1 infusion in rats was shown to increase renal blood flow, eGFR, and urine output in a PKA-dependent manner. In this model, the effect was mediated by the phosphorylation of a Na+/H+ transporter NHE3 at PKA consensus sites, causing its inhibition [78]. Others have confirmed these findings and further concluded that exenatide-induced diuresis via PKA-dependent phosphorylation of NHE3 causes a decrease in blood pressure in the db/db mouse model [79]. Combined, these preclinical studies point to the mechanisms behind the beneficial effects of GLP1RA seen in clinical trials on kidney disease.

## 7. GLP1RA Effects on Cancer

In the U.S., cancer is the seventh leading cause of death, and the incidence of most cancers increases with age, with the population older than 55 accounting for 78% of all new diagnoses [80]. Interpreting the effects of GLP1RA on cancer initiation and progression is complicated due to the diversity of malignant diseases. Differences in the expression of the GLP1R in different tumors or differential expression between healthy and diseased tissue could partially explain this differential effect on different cancers. A study on the expression of the GLP1R in cancerous and normal tissues showed that the GLP1R had the highest expression in endocrine tumors. It also showed expression in embryonic tumors, tumors of the nervous system, and some carcinomas [81]. In normal tissues resected at the same time as the tumor, expression was found in the central nervous system, pancreas, small and large intestines, breast, lung, and kidney tissues [81]. The effects of GLP1A seem to depend on the specific tumor type being studied. For this reason, the benefit of GLP1RA on cancer outcomes and their beneficial effects on other diseases of aging must be weighed against the potential increased risk of some cancers.

Semaglutide has a boxed warning from the FDA about its potential for causing thyroid C-cell tumors, specifically medullary thyroid carcinoma [82]. Subsequent clinical data have confirmed this finding. A study of the FDA adverse event reporting system identified an increased risk of thyroid cancer in GLP1RA-treated patients when compared to patients on metformin; however, the increased rate of thyroid cancer was not significant in the liraglutide-treated groups, showing that the cancer association may be drug-dependent and not a feature of the entire class of drugs [83]. A similar study of this database found that GLP1RA usage was associated with increased incidence of not only thyroid cancers but also pancreatic and neuroendocrine neoplasms [84]. Semaglutide does have a warning label about the risk of pancreatitis, although it does not mention the risk of pancreatic cancer [82]. Pancreatitis is a risk factor for pancreatic cancer, and an increased risk of pancreatic cancer with GLP1RA usage has been shown in some studies [85]. 

On the other hand, a meta-analysis of randomized clinical trials and a systematic review of observational studies reported no increased risk of pancreatitis or pancreatic cancer in patients treated with GLP1RA [86,87]. Moreover, for many malignancies, there is evidence of an association between GLP1R activation and reduced cancer incidence. Increased fasting GLP1 plasma concentrations are associated with decreased first incidence of cancer [88]. Although GLP1RA often cause super-physiological levels of GLP1R activation, this study provides evidence that GLP1R signaling may decrease cancer risk. In a review of randomized control trial data, liraglutide was shown to decrease the risk of prostate cancer. However, no significant positive or negative relationship was found with any other cancer type [89]. A separate study of the FDA adverse event reporting system found that GLP1RA usage was linked with a significant decrease in not only prostate cancer but also colon and lung cancer [83]. The differences in outcomes from these studies could be explained by differences in length of follow-up, type of GLP1RA used, and characteristics of the study populations. Considering these mixed data, it is crucial to better understand the role of GLP1R agonism in cancer as these drugs become more widely adopted. 

In support of the possible benefits of GLP1RA for patients with cancer, it has been reported that exenatide inhibits the proliferation of prostate cancer cells. This effect is attenuated by GLP1R knockdown or treatment with a GLP1R antagonist. In the same study, researchers showed in vivo that exenatide treatment caused a significant decrease in prostate tumor size with no effect on body weight or glucose levels [90]. Another study of exenatide in prostate cancer showed its ability to cause cell cycle arrest in vitro in an AMPK-dependent manner and to sensitize prostate cancer to ionizing radiation in vitro and in vivo [91]. The preclinical evidence on the benefits of GLP1R in other types of cancer is sparse and mixed. One group has reported that exenatide inhibits proliferation and induces apoptosis of the mouse CT26 colon cancer cell line and increases their susceptibility to topoisomerase inhibitor irinotecan. In vivo, exenatide treatment of orthotopic CT26 tumors showed no change in tumor weight; however, these tumors had increased TUNEL staining and decreased BrdU incorporation, indicating increased intra-tumoral apoptosis and decreased cell proliferation. Despite these positive findings for CT26 cells, GLP1R expression was not observed in any other human or mouse colon cancer cell line tested [92]. A subsequent study found that exenatide only had a growth inhibitory effect on CT26 cells and did not affect cell death, migration, or adhesion in vitro or tumor growth in vivo for any other colon cancer cell line tested [93]. Nevertheless, liraglutide has been shown to decrease tumor size in orthotopic models of lung cancer and liver cancer, and it increased the sensitivity of both tumors to immune checkpoint blockade. After surgical resection of the primary tumor, orthotopic transplant of the same cancer cell line on the opposite side had decreased tumor growth, demonstrating a possible sustained anti-tumor effect of GLP1RA [94]. These studies suggest the possibility that the beneficial effects of GLP1RA in cancer are mediated by their effects on non-malignant cells in the tumor microenvironment.

It is important to note that, like the other diseases discussed in this review, metabolic syndrome is a risk factor for many types of cancer, namely, that of the liver, pancreas, colon, and breast [80]. Therefore, the anti-cancer effects of GLP1RA could be an indirect result of its glucose control and weight management (Figure 1). However, lung and prostate cancers, which have been found to be suppressed by GLP1RA in mouse models, are not strongly associated with diabetes [80]. This finding indicates that the anti-cancer effects of GLP1RA are, at least in these cases, mediated by direct effects independent of their effects on metabolism. The conflicting data on the role of GLP1RA in human cancers could be due to several factors, including the type of tumor, the design of the clinical trials, the specific GLP1RA used, or the population being studied. The increasing use of GLP1RA in populations with cancer and at risk for cancer development will help better define their potential benefits for individual cancer types.

## 8. GLP1RA Effects on Cellular Aging

Aging is unavoidable; however, there is an important distinction between chronological and biological age. Biological age can be altered by diet, physical activity, and various environmental factors. With the growing aging population, it is crucial to elucidate the mechanisms underlying healthy aging and develop treatments that augment these mechanisms. Biological aging is driven by multiple factors, including telomere shortening, epigenetic changes, stem cell exhaustion, mitochondrial dysfunction, genome instability, impaired protein metabolism, and abnormal nutrient sensing. All these cellular changes can induce apoptosis or cellular senescence, a state characterized by irreversible cell cycle arrest, a proinflammatory senescence-associated secretory phenotype (SASP), and expression of senescence markers [95,96]. Accumulation of senescent cells is responsible for tissue changes leading to aging-related diseases [97,98]. Because cell senescence contributes to biological aging [99], approaches to suppress or reverse senescence would be integral in preventing and treating aging-associated diseases [100]. 

Evidence from preclinical studies has suggested that GLP1R activation can suppress some cellular changes that accompany aging. It has been shown that the shortening of telomeres and a decline in DNA repair mechanisms induce cell senescence and aging [100]. In fact, mutations in DNA repair genes are responsible for Werner Syndrome and Hutchinson–Gilford progeria syndrome, the diseases of accelerated aging [99]. It was recently reported in non-human primates that plasma GLP1 levels in adolescents were positively correlated with their adult telomere length, showing a possible association between GLP1R activity and telomere maintenance [101]. Base excision repair is the primary pathway of DNA damage repair in neurons. The GLP1RA exenatide ameliorates drug-induced ROS production in neurons by increasing expression of the base excision repair protein, APE1, and activating this geno-protective pathway in vitro and in vivo [102]. Exenatide was also shown to decrease the number of senescent endothelial cells and improve their survival after a hydrogen peroxide challenge by modulating signaling through Sirt1, PPARγ, and PGC1α [103]. Moreover, exenatide can alleviate angiotensin II-mediated senescence in VSMC [47]. In this study, the decrease in cell senescence was accompanied by improved proliferative capacity and decreased oxidative stress. This body of evidence suggests that GLP1RA may be able to delay aging through a multi-pronged mechanism suppressing cell senescence and maintaining genomic integrity. 

Caloric restriction is an intervention that has consistently been shown to extend lifespan across organisms. With caloric restriction, there are intracellular changes in energy-sensing pathways that integrate the metabolic state and alter signaling pathways. mTOR and AMPK signaling are the key nutrient-sensing pathways that mediate the beneficial effects of caloric restriction on longevity [99]. It has been shown that activation of AMPK increases the lifespan [104,105]. In a study of diabetic rats, the GLP1RA exenatide decreased cardiomyocyte apoptosis and improved cardiac function. This effect was associated with increased AMPK phosphorylation, activating its activity [106]. In addition, it has been reported that liraglutide ameliorates the lipotoxic, inflammatory effects of high-calorie diet feeding in vivo and palmitic acid treatment in vitro via inhibition of the mTORC1 signaling pathway [107]. In another study, in vitro treatment of endothelial cells with liraglutide caused mTORC2-dependent AKT phosphorylation and release of nitric oxide (NO). Importantly, this study provides evidence of a liraglutide-mediated increase in telomerase activity and its translocation to the nucleus [108]. Telomerase not only lengthens telomeres to prevent their attrition [109,110] but is also implicated in protection from genotoxic stress [111] and has genome-wide telomere-independent effects on gene expression and cell metabolism [112]. There is accumulating evidence that inactivation of telomerase is a key contributor to cell aging, leading to senescence [113]. We have reported that telomerase inactivation in adipose progenitor cells causes cellular senescence and predisposes mice to AT dysfunction and T2D [114]. Conversely, telomerase gene therapy delays aging and increases longevity without increasing cancer incidence in mice [115]. It is tempting to speculate that the effect of GLP1RA on telomerase and nutrient sensing pathways may account for their beneficial role in diseases of aging. However, it is apparent that healthy aging can also be promoted by the positive effects of GLP1RA on multiple individual organs that collectively improve physiological homeostasis (Figure 1).

## 9. Conclusions 

The widespread adoption of GLP1RA has been instrumental in controlling glucose metabolism and supporting adiposity management in diabetic and overweight patients. Clinical trials have revealed that GLP1RA signaling spans beyond the pancreas and brain, potentially explaining their beneficial effects in other organs (Table 1, Table 2, Table 3 and Table 4). GLP1RA appear to suppress aging-related diseases in the cardiovascular system, kidney, and other organs (Figure 1). Diabetes and obesity both predispose patients to cardiovascular diseases, neurodegenerative diseases, and many other diseases of aging. Therefore, some of the benefits from GLP1RA therapy are likely to be secondary to the improvement in whole-body glucose metabolism through effects on the pancreas and adipose tissue control through effects on the central nervous system. However, accumulating evidence from preclinical models indicates that GLP1RA directly signal in multiple organs by inducing cellular pathways that prevent pathogenesis and/or suppress disease progression. Going forward, it will be important to better understand the organ-specific and cell type-specific roles of GLP1RA and their possible effects, likely synergizing with the benefits of GLP1RA on metabolism and body composition. 

With the widespread beneficial effects of GLP1RA on the body, many regard them as miracle drugs. As newer anti-diabetic therapies come to market, it will be important to evaluate their systemic effects in the context of the anti-aging benefits of GLP1RA discussed in this review. The newly developed dual agonists, which activate both GLP1R and glucose-dependent insulinotropic peptide (GIP) receptor (GIPR), appear more efficacious than GLP1RA alone in reducing HbA1C and body weight [116]. Dual agonists have been extensively reviewed elsewhere [117,118,119,120,121] and tirzepatide (Zepbound) was recently FDA-approved. Like the GLP1RA, dual agonists have shown efficacy in aging-related diseases besides diabetes and obesity including cardiovascular disease [119,120,122], neurodegenerative diseases [123,124,125], and kidney disease [126]. In the future, it will be important to determine if/how the addition of GIP agonism affects the anti-aging effects of GLP1RA. Ultimately, expanding the indications for GLP1RA, and possibly dual agonists, beyond diabetes and obesity could improve health and prolong healthy aging in a more general population. 

## Figures and Tables

**Figure 1 cells-13-00065-f001:**
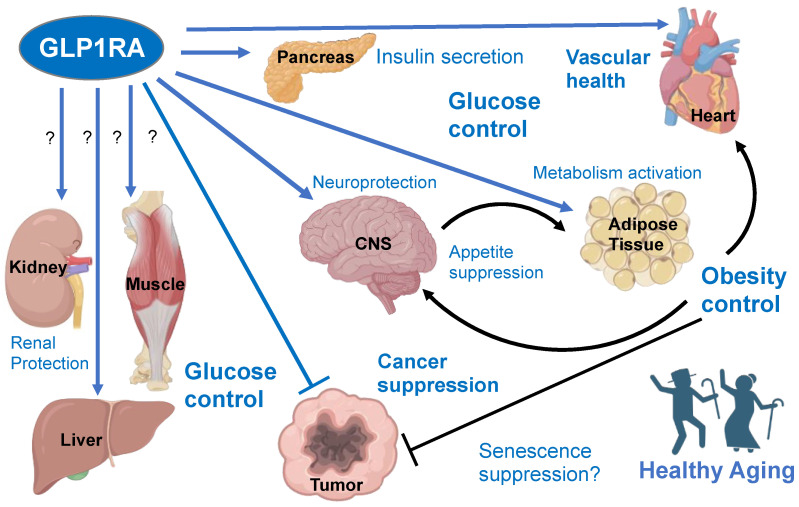
The dichotomous effects of GLP1 receptor agonists (GLP1RA). Effects are labeled with blue font, with resulting physiological changes marked in bold. GLP1R is expressed in some but not all cell populations of organs shown. Direct signaling reported to have positive effects is indicated by blue connections. Effects unclear from literature or remaining to be tested/confirmed are marked with “?”. Indirect effects downstream of GLP1RA signaling are shown as black connections. GLP1RA have also been shown to decrease inflammation in multiple organs, which also likely contributes to their indirect beneficial effects.

**Table 1 cells-13-00065-t001:** Summary of clinical trials showing the effects of GLP1RA in Sarcopenia.

Drug	Population	Outcomes	Reference
GLP-1	Men ≥ 70	Increased muscle protein synthesis and muscle microvascular blood flow; Decreased muscle protein breakdown	[12]
Dulaglutide	Type 2 Diabetes on Hemodialysis	Decreased fat mass and skeletal muscle mass	[13]
Liraglutide	Overweight and Obese with Type 2 Diabetes	Increased skeletal muscle index	[14]
Liraglutide	Overweight and Obese with Type 2 Diabetes	No change in relative skeletal muscle mass	[15]
Exenatide	Obese with Type 2 Diabetes	No change in muscle mass	[16]
GLP1	Obesity	Increased microvascular blood flow in cardiac and skeletal muscle	[17]

**Table 2 cells-13-00065-t002:** Summary of clinical trials showing the effects of GLP1RA in cardiovascular diseases.

Drug	Population	Outcomes	Reference
Liraglutide	Type 2 Diabetes with High CV Risk	Decreased time to first occurrence of nonfatal MI or nonfatal stroke or death due to CV event; Fewer occurrences of death due to CV event; Decreased all-cause mortality	[48]
Semaglutide	Type 2 Diabetes	Decreased risk of nonfatal stroke; No change in nonfatal MI or death from CV events	[49]
Dulaglutide	Type 2 Diabetes with Previous CV Event or CV Risk Factors	Decreased first occurrence of nonfatal stroke; No change in all-cause mortality; No change in first occurrence of nonfatal MI or death from CV event	[50]
Albiglutide	Type 2 Diabetes with CV Disease	Decreased risk of MI; No change in stroke risk, death from CV event, or all-cause mortality	[51]
Exenatide	Type 2 Diabetes	Decreased death from any cause; No change in death from CV events, MI, stroke, HF hospitalizations, or ACS hospitalizations	[52]
Lixisenatide	Type 2 Diabetes with Recent Acute Coronary Event	No change in nonfatal MI, nonfatal stroke, unstable angina, HF hospitalizations, or all-cause mortality	[53]
Liraglutide	HF or HF with Reduced Ejection Fraction	No change in all-cause mortality, HF hospitalizations, death due to HF, or NT-proBNP	[54]
Efpeglenatide	Type 2 Diabetes with High CVD Risk	Decreased HF; No change in MI, stroke, CVD mortality, total mortality, unstable angina, coronary revascularization	[55]

CV: cardiovascular, CVD: cardiovascular disease, MI: myocardial infarction, HF: heart failure, ACS: acute coronary syndrome, NT-proBNP: N terminal-prohormone of brain natriuretic peptide.

**Table 3 cells-13-00065-t003:** Summary of clinical trials showing the effects of GLP1RA in neurodegenerative diseases.

Drug	Population	Outcomes	Reference
Exenatide	High Risk for AD	No change in cognitive measures, MRI cortical thickness or volume, or CSF biomarkers; Reduced amyloid beta in plasma extracellular vesicles	[61]
Exenatide	Moderate Parkinson’s Disease	Improvement in MDS-UPDRS compared to unmedicated patients; No difference in MDS-UPDRS compared to SOC medication; Decreased rate of decline in DaT scan	[62]
Liraglutide	Mild to Moderate AD	Improved temporal lobe and cortical MRI volume; Improved cognitive function	[63]
Mixed GLP1RA	Nationwide cohort of Type 2 Diabetes	Lower rates of dementia	[58]
Mixed GLP1RA	Pooled RCT Data	Lower rates of developing dementia	[58]
Dulaglutide	Analysis of RCT of Type 2 Diabetes with Previous CV Event or CV Risk Factors	Smaller decline in cognitive ability	[60]

AD: Alzheimer’s Disease, MRI: magnetic resonance imaging, CSF: cerebrospinal fluid, MDS-UPDRS: Movement Disorder Society Unified Parkinson’s Disease Rating Scale, SOC: standard of care, DaT: dopamine active transporter.

**Table 4 cells-13-00065-t004:** Summary of clinical trials showing the effects of GLP1RA in renal disease.

Drug	Population	Outcomes	Reference
Efpeglenatide	Type 2 Diabetes with a history of CVD	Decreased rate of incident macroalbuminuria	[55]
Dulaglutide	Type 2 Diabetes	Decreased incidence of new macroalbuminuria and sustained eGFR decline ≥ 40%; No change in CRRT	[71]
Liraglutide	Type 2 Diabetes with Diabetic Nephropathy	Decreased proteinuria and rate of decline in eGFR	[74]
Liraglutide	Type 2 Diabetes	Reversible decrease in eGFR; Decreased SBP, pro-ANP, and UACR	[75]
Dulaglutide	Type 2 Diabetes with Moderate to Severe CKD	eGFR higher than insulin-treated patients	[70]

eGFR: estimated glomerular filtration rate, CRRT: continuous renal replacement therapy, SBP: systolic blood pressure, pro-ANP: pro-atrial natriuretic peptide, UACR: urinary albumin to creatinine ratio, CKD: chronic kidney disease.

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
