# Peer review of "GLP1 Receptor Agonists—Effects beyond Obesity and Diabetes"

_cells, 2023, doi:10.3390/cells13010065_

Round 1

Reviewer 1 Report

Comments and Suggestions for Authors

In the present review the Authors summarize the studies reporting the effects of GLP1RA on aging-related diseases and discuss potential underlying mechanisms.

Overall, the manuscript looks weak to be accepted in this journal. This paper is an elegant narrative review, complete and well-designed. However, a few concerns/comments needed to be explained/modified.

1.      The benefits of GLP1-RA to reduce cardio-renal disease were at least as large among people with estimated glomerular filtration rate <60 mL/min/1.73 m2. GLP1-RA also conferred a 21% reduction in kidney outcomes, largely through reduction in albuminuria. It remains uncertain whether GLP1-RA would confer similar favorable results for eGFR decline and/or progression to end-stage kidney disease. The Authors included some information in the introduction section, but it would be interesting a new paragraph on GLP1RA Effects on kidney protection, including data from both clinical (Zeng ZY, Yang CT, Lin WH, Yao WY, Ou HT, Kuo S. Chronic kidney outcomes associated with GLP-1 receptor agonists versus long-acting insulins among type 2 diabetes patients requiring intensive glycemic control: a nationwide cohort study. Cardiovasc Diabetol. 2023 Oct 4;22(1):272. doi: 10.1186/s12933-023-01991-5; Meng Y, Wang Y, Fu W, Zhang M, Huang J, Wu H, Sun L. Global trends and focuses of GLP-1RA in renal disease: a bibliometric analysis and visualization from 2005 to 2022. Naunyn Schmiedebergs Arch Pharmacol. 2023 Dec;396(12):3347-3361. doi: 10.1007/s00210-023-02575-6. Epub 2023 Jun 30. PMID: 37389601) and pre-clinic data (GLP-1RAs inhibit the activation of the NLRP3 inflammasome signaling pathway to regulate mouse renal podocyte pyroptosis. doi: 10.1007/s00592-023-02184-y; Michos ED, Bakris GL, Rodbard HW, Tuttle KR. Glucagon-like peptide-1 receptor agonists in diabetic kidney disease: A review of their kidney and heart protection. Am J Prev Cardiol. 2023 May 24;14:100502. doi: 10.1016/j.ajpc.2023.100502)

2.      Maybe kidney protection and its pathways have to be included also in Figure 1.

3.      Table 3 and Table 4 appear in the text before Table 1 and Table 2

Comments on the Quality of English Language

Minor editing of English language required

Author Response

We are pleased with the positive feedback and are grateful for the constructive criticisms and suggestions. We have revised the manuscript according to them. 
  1. The benefits of GLP1-RA to reduce cardio-renal disease were at least as large among people with estimated glomerular filtration rate <60 mL/min/1.73 m2. GLP1-RA also conferred a 21% reduction in kidney outcomes, largely through reduction in albuminuria. It remains uncertain whether GLP1-RA would confer similar favorable results for eGFR decline and/or progression to end-stage kidney disease. The Authors included some information in the introduction section, but it would be interesting a new paragraph on GLP1RA Effects on kidney protection, including data from both clinical (Zeng ZY, Yang CT, Lin WH, Yao WY, Ou HT, Kuo S. Chronic kidney outcomes associated with GLP-1 receptor agonists versus long-acting insulins among type 2 diabetes patients requiring intensive glycemic control: a nationwide cohort study. Cardiovasc Diabetol. 2023 Oct 4;22(1):272. doi: 10.1186/s12933-023-01991-5; Meng Y, Wang Y, Fu W, Zhang M, Huang J, Wu H, Sun L. Global trends and focuses of GLP-1RA in renal disease: a bibliometric analysis and visualization from 2005 to 2022. Naunyn Schmiedebergs Arch Pharmacol. 2023 Dec;396(12):3347-3361. doi: 10.1007/s00210-023-02575-6. Epub 2023 Jun 30. PMID: 37389601) and pre-clinic data (GLP-1RAs inhibit the activation of the NLRP3 inflammasome signaling pathway to regulate mouse renal podocyte pyroptosis. doi: 10.1007/s00592-023-02184-y; Michos ED, Bakris GL, Rodbard HW, Tuttle KR. Glucagon-like peptide-1 receptor agonists in diabetic kidney disease: A review of their kidney and heart protection. Am J Prev Cardiol. 2023 May 24;14:100502. doi: 10.1016/j.ajpc.2023.100502).

The new paragraph on GLP1RA effects on kidney protection is now included with new references.

  1. Maybe kidney protection and its pathways have to be included also in Figure 1.

Kidney protection is now integrated into Figure 1.

  1. Table 3 and Table 4 appear in the text before Table 1 and Table 2

The renal outcome clinical trial table is now modified and the numbering of tables is fixed.

Reviewer 2 Report

Comments and Suggestions for Authors

The manuscript entitled "GLP1 receptor agonists - effects beyond obesity and diabetes" by Wilbon SS and Kolonin MG reviewed the benefits of the GLP1RA administration to reduce hyperglycaemia, body weight as well as cardiovascular complications, neurodegeneration, cancer and ageing. The paper is well structured. 

The strength of the manuscript is given by the importance of the topic, given the ongoing interest and research characterizing new GLP1 derivatives.

The manuscript nicely correlated clinical with pre-clinical studies for each of the six pathologies presented. However, in the conclusion the authors suddenly suggested the association of GLP1RA with GIP for therapy, without previously introducing GIP to the reader. Therefore, I recommend the authors to include a small chapter before conclusion, to discuss and bring evidence about the additional benefit obtained of expected following the combination of GLP1RA with other drugs such as GIP or metformin, also in line with the pathologies described at chapters 2-6 and ageing. 

Author Response

We are pleased with the positive feedback and are grateful for the  suggestion. We found it necessary to mention GIPRA because Trizepatide is important to acknowledge and because it contributes to GLP1R efficacy. GIP and metformin (mentioned earlier in the text) cannot be covered in depth in this short review focusing on GLP1RA. However, we expanded the paragraph to emphasize the importance of considering the effects of dual agonists in the context of other pharmacological metabolism interventions.